# FAK Inhibitor-Based Combinations with MEK or PKC Inhibitors Trigger Synergistic Antitumor Effects in Uveal Melanoma

**DOI:** 10.3390/cancers15082280

**Published:** 2023-04-13

**Authors:** Malcy Tarin, Fariba Némati, Didier Decaudin, Christine Canbezdi, Benjamin Marande, Lisseth Silva, Héloïse Derrien, Aart G. Jochemsen, Sophie Gardrat, Sophie Piperno-Neumann, Manuel Rodrigues, Pascale Mariani, Nathalie Cassoux, Marc-Henri Stern, Sergio Roman-Roman, Samar Alsafadi

**Affiliations:** 1Translational Research Department, Institut Curie, PSL Research University, 75005 Paris, France; 2Laboratory of Preclinical Investigation, Institut Curie, PSL Research University, 75005 Paris, France; 3Department of Medical Oncology, Institut Curie, PSL Research University, 75005 Paris, France; 4Department of Cell and Chemical Biology, Leiden University Medical Center, 2300 RC Leiden, The Netherlands; 5Department of Biopathology, Institut Curie, PSL Research University, 75005 Paris, France; 6INSERM U830, DNA Repair and Uveal Melanoma, Institut Curie, PSL Research University, 75005 Paris, France; 7Department of Ocular Oncology, Institut Curie, Université Paris Cité, 94010 Paris, France

**Keywords:** metastatic uveal melanoma, combination treatments, FAK inhibition

## Abstract

**Simple Summary:**

Uveal Melanoma (UM) is a rare and malignant intraocular tumor with dismal prognosis. Despite efficient control of the primary tumor by radiation or surgery, up to 50% of patients subsequently develop metastases, mainly in the liver. The treatment of UM metastases is challenging and the patient survival is very poor. Today, most of the tested treatments including inhibitors of protein kinase C (PKC), mitogen-activated protein kinase (MEK) or immune checkpoint blockade have been largely ineffective in patients with metastatic UM. Given that single-agent targeted therapies often activate compensatory mechanisms, combination strategies are relevant to evaluate in UM in preclinical and clinical settings. Our study confirms the previously described synergy of the dual inhibition of focal adhesion kinase (FAK) and MEK, and identifies a novel combination of drugs (FAK and PKC inhibitors) as a promising strategy for therapeutic intervention in metastatic UM.

**Abstract:**

Uveal Melanoma (UM) is a rare and malignant intraocular tumor with dismal prognosis. Even if radiation or surgery permit an efficient control of the primary tumor, up to 50% of patients subsequently develop metastases, mainly in the liver. The treatment of UM metastases is challenging and the patient survival is very poor. The most recurrent event in UM is the activation of Gαq signaling induced by mutations in GNAQ/11. These mutations activate downstream effectors including protein kinase C (PKC) and mitogen-activated protein kinases (MAPK). Clinical trials with inhibitors of these targets have not demonstrated a survival benefit for patients with UM metastasis. Recently, it has been shown that GNAQ promotes YAP activation through the focal adhesion kinase (FAK). Pharmacological inhibition of MEK and FAK showed remarkable synergistic growth-inhibitory effects in UM both in vitro and in vivo. In this study, we have evaluated the synergy of the FAK inhibitor with a series of inhibitors targeting recognized UM deregulated pathways in a panel of cell lines. The combined inhibition of FAK and MEK or PKC had highly synergistic effects by reducing cell viability and inducing apoptosis. Furthermore, we demonstrated that these combinations exert a remarkable in vivo activity in UM patient-derived xenografts. Our study confirms the previously described synergy of the dual inhibition of FAK and MEK and identifies a novel combination of drugs (FAK and PKC inhibitors) as a promising strategy for therapeutic intervention in metastatic UM.

## 1. Introduction

Uveal melanoma (UM) is the most common malignant intraocular tumor in adults, affecting five individuals per million per year in the Caucasian population [1]. Although primary UM is controlled by irradiation or enucleation, up to 50% of UM patients develop metastases (mUM)—mainly in the liver—which are associated with poor prognosis and a median survival of 12 months [2]. Thus, there is an urgent need to develop new therapeutic options for mUM [3,4].

In contrast to cutaneous melanoma, UM presents a low mutational burden with two driver events per tumor, as well as few genomic alterations (chromosome loss of 1p, 3, 6q and 8p and gain of 8q) [5]. More than 90% of UM harbor constitutively active mutations in GNAQ and GNA11, encoding for the alpha subunits Gαq and Gα11 of the heterotrimeric G protein [6,7,8]. An additional 10% of UM carry mutations in the Gαq-linked receptor CYSLTR2 [9], or in PLCB4 encoding phospholipase C β4, the immediate downstream enzyme of Gαq [10]. These driver mutations lead to the activation of downstream effectors including protein kinase C (PKC) and mitogen-activated protein kinases (MAPK), implying a strong rationale for the therapeutic targeting of these activated pathways in UM [4]. Gαq pathway-related mutations are thought to arise early during tumor evolution as they are already found in benign lesions [6,11]. A second event consisting of mutually exclusive mutations in BAP1, SF3B1, or EIF1AX genes is required for the full malignant transformation of uveal melanocytes to UM [12,13,14,15]. 

It has been shown recently that activating mutations in GNAQ/11 trigger not only the canonical pathway leading to the activation of the PKC-MAPK cascade but also a non-canonical pathway leading to the Hippo/YAP pathway activation via TRIO-RhoA-and Focal adhesion kinase (FAK) [16,17]. FAK has been described as an integral node of this non-canonical Gαq pathway [18]. Agents targeting the Gαq canonical signaling pathway, PLCβ–PKC–MAPK, have been shown to have nearly no impact on the overall survival of patients with mUM, as single agents or when combined with chemotherapy [19,20,21,22].

In fact, most of the tested treatments including inhibitors of PKC, MEK or immune checkpoint blockade have been largely ineffective in patients with mUM [20,23]. Very recently, bispecific fusion protein-based treatment (Tebentafusp) resulted in longer overall survival in HLA-A02:01-positive patients with mUM. Yet, this treatment concerns only half of UM Caucasian patients, and questions remain regarding the duration of treatment in the absence of objective response in the majority of patients [24].

Given that single-agent targeted therapies often activate compensatory mechanisms, combination strategies are relevant to evaluate in UM in preclinical and clinical settings. Recently, a combination of FAK and MEK inhibitors has been shown to synergize both on in vitro and in vivo UM models [17]. Following these observations, a clinical trial has been started to evaluate the potential effect of combinations of FAK and RAF/MEK inhibitors [25,26].

Here, we used our previously established pipeline of drug combination screening in UM cell lines to identify new potentially synergistic combinations [27]. We evaluated FAK inhibitor-based combinations inspired by the latest molecular characterization findings in UM. We found that FAK inhibitor-based combinations with MEK or PKC inhibitors are highly synergistic in UM cell lines. The synergistic potential of these two combinations was further confirmed in UM patient-derived xenografts (PDXs).

Overall, we confirm the synergistic effect previously observed by combining FAK and MEK inhibitors [17], and we identify the combination of FAK and PKC inhibitors as a very promising therapeutic strategy that should be considered in clinical trials as a potential treatment for mUM.

## 2. Materials and Methods

### 2.1. Cell Culture 

MP38, MP46, MP65, MM28 and MM66 were established in our laboratory [28]. OMM1, OMM2.3 and OMM2.5 were kindly provided by P.A. van der Velden (Leiden University Medical Center, Leiden, The Netherlands). The main characteristics of the used cell lines and all culture conditions are described in Appendix A.

### 2.2. Compounds 

All drugs and inhibitors (Appendix A) were purchased from Selleckchem for in vitro, and MedChemExpress for in vivo studies. For the in vitro study, all drugs were dissolved in dimethyl sulfoxide (DMSO) at 10 mM and stored at −20 °C. Further dilutions were made according to each experimental design. 

### 2.3. Drug Combination Cell Viability Screen

Cells were seeded at appropriate concentration in 96-well plates. The day after, each drug was added following a 6 × 6 matrix dilution design, as described in a previous paper and in Appendix A [27,29]. Cell viability was measured after 5 days of treatment using MTT assay (Sigma, St. Quentin Fallavier, France). For all different combinations, three technical replicates and three independent biological replicates were performed on whole cell line panel. Plates reading, results analysis and determination of the combination’s synergistic potential were done, as detailed in a previous paper [27,29].

### 2.4. Caspase-3/7 Activity Assay

According to the screening, cells were treated with different compounds alone or in combinations at synergistic doses. Caspase-3/7 activity was determined by using a fluorescence-based assay (Caspase-Glo 3/7 Assay, Promega, Madison, WI, USA). The assay was performed according to the manufacturer’s recommendations after 3 and 5 days of treatment. 

### 2.5. Immunoblotting Analyses

Cells were cultured in 10 cm-diameter dishes and treated with DMSO or each drug alone or in combination for 1 and 2 h, or for 3 days and 5 days, at synergistic doses. After separation by SDS-PAGE and transferring onto polyvinylidene difluoride (PVDF) mem-branes, protein extracts were probed with antibodies against pFAK (#8556), FAK (#3285), pMARCKS (#11992), MARCKS (#5607), pERK (#9101), ERK (#9102), cleaved and total PARP (#9542), and GAPDH (#97166), all purchased from Cell Signaling Technology, Danvers, MA, USA. GAPDH immunoblotting was used to quantify and normalize results. Immunolabelled proteins were detected using Odyssey secondary antibodies coupled to a 700 or 800 nm and the Odyssey Infrared Imaging System (Li-cor). 

### 2.6. UM PDX Models

Three PDXs derived from liver metastases of patients, MM26, MM309 and MM339 were used for this study. The main molecular features of these models are presented in the Appendix A. All these models have been obtained from liver metastases of UM patients, with consent and ethics approval of our establishment. The in vivo methodology used for the establishment of these PDXs has previously been reported [30].

### 2.7. In Vivo Tumor Growth and Antitumor Efficacy

The experimental protocol of the in vivo study is detailed in the Appendix A [31,32]. Female SCID mice were used (Charles River Laboratories, Miserey, France). Animal care and use for this study were performed in accordance with the recommendations of the European Community (2010/63/UE) for the care and use of laboratory animals. Experimental procedures were specifically approved by the ethics committee of the Institut Curie CEEA-IC #118 (Authorization APAFiS #25870-2020060410487032-v1 given by National Authority) in compliance with the international guidelines.

The mice were xenografted with a tumor fragment of 20-40 mm^3^. Mice bearing growing tumors with a volume of 60–150 mm^3^ were randomly assigned to the control or treatment groups; 4 to 10 mice were included per group. Treatments were started on day one. The doses, routes of administration, and schedules of drugs (VS4718, LXS196, and Trametinib) are shown in Appendix A. Two different schedules were assessed to evaluate anti-tumor efficacy with an administration four or five days per week every week (continuous schedule) or every two weeks (discontinuous schedule). Mice were weighed and tumors were measured twice a week. Tumor volumes were calculated by measuring two perpendicular diameters with a caliper. Individual tumor volume, relative tumor volume (RTV), and tumor growth inhibition (TGI) were calculated according to standard methodology [30]. Similarly, in order to evaluate all therapies and combinations according to individual mouse variability, the overall response rate (ORR) to treatments and the probability of progression were calculated as previously reported [33]. Protein extraction from frozen tumor samples was realized as described previously [34], and protein expression analysis was done in the same manner as for the in vitro assay.

## 3. Results

### 3.1. Identification of Synergistic FAK Inhibitor-Based Combinations in Uveal Melanoma Cell Lines

Given that FAK has recently been identified as a downstream target of Gαq signaling [17], which is constitutively activated in UM, we aimed to identify potential FAK inhibitor-based drug combinations susceptible to synergistically reduce the viability of UM cell lines.

We assessed dual drug combinations in a panel of eight UM cell lines derived either from metastases or from primary tumors harboring the most frequent genetic features found in mUM (i.e., GNAQ or GNA11 mutations associated with BAP1 deficiency). We evaluated combinations of the FAK inhibitor VS4718 with inhibitors of PKC (LXS196), MEK (Trametinib), MDM2 (HDM201), BCL2 and BCL2/xL (ABT199 and ABT263, respectively), which are the best-ranked inhibitors previously reported in UM cell models [27]. All combinations were assessed for synergy based on cell viability and according to the Bliss independence model. Figure 1 shows the ranking of the tested combinations according to their median Best Excess Over Bliss values, including those of our previously established pipeline as reference values. Our findings indicate that the VS4718 highly synergizes with LXS196 and Trametinib, and these two combinations ranked in the top five out of forty-two dual drug combinations. As depicted in Appendix A, the response to single-agent treatments tends to reach a residual threshold in most UM cell lines, suggesting activation of compensatory resistance mechanisms in the residual cells, which also provides a rationale for combined treatment. An exquisite sensitivity to LXS196 was observed in MM66 cells (Appendix A).

### 3.2. Apoptosis Induction after Treatment with VS4718 Combined with LXS196 or with Trametinib

We evaluated apoptosis induction under treatment with the two most synergistic FAK inhibitor-based combinations (VS4718 with LXS196 or Trametinib) in four UM cell lines: two derived from mUM patients (OMM1 and MM66) and two derived from primary UM patients (MP38 and MP65). Apoptosis was assessed based on caspase 3/7 activity after 3 and 5 days of treatment. 

Single agent treatment did not lead to any significant activation of caspase 3/7, an indicator of apoptosis induction, in OMM1, MP38 or MP65 (Figure 2). LXS196 alone significantly increased caspase 3/7 activity in MM66. Of note, this response of MM66 to LXS196 alone was also observed in terms of cell viability, suggesting an exceptional sensitivity of this cell line to LXS196 (Appendix A). After 5 days of treatment with LXS196 alone, we observed apoptosis induction in MP38, MM66, and MP65 cells. The combined treatment of VS4718 with LXS196 or with Trametinib induced a significant increase of apoptosis induction as compared to single agent treatments except for OMM1 cell line. This increase of caspase 3/7 activation was more significant at 5 days of treatment. Interestingly, VS4718 combined with LXS196 had a significantly higher synergistic effect compared to VS4718 combined with Trametinib in MP65 and MM66 cells.

Increased apoptosis induction with FAK inhibitor-based combinations supports the synergistic effects previously observed on cell viability, and highlights the therapeutic potential of the combination of FAK inhibitors with PKC or MEK inhibitors.

### 3.3. Validation of Target Engagement under Treatment with FAK Inhibitor Drug Combinations

We examined the engagement of known deregulated signaling pathways in UM by protein expression analysis at early time points (1 and 2 h) after treatment with single agents or combinations in the four selected UM cell lines. 

As shown in Figure 3, VS4718 reduced phosphorylated FAK, Trametinib reduced phosphorylated ERK, while LXS196 decreased phosphorylated MARCKS levels in all cell lines, validating the engagement of the target for each inhibitor in these treatment conditions. It has been recently reported that the PKC inhibitor LXS196 reduces the phosphorylation of FAK in UM cell lines, suggesting that PKC controls FAK activation [35]. In our experiments, LXS196 at a concentration (500 nM) which significantly diminishes MARCKS phosphorylation does not affect the phosphorylation of FAK in the four tested UM cell lines. We could not observe any significant effect of LXS196 on FAK phosphorylation in a dose-response experiment using the same UM cell lines (MP41 and Mel202) described elsewhere [35]. Our data suggest that the activation of FAK in UM cells is not PKC-dependent. At late time points (3 and 5 days), we also confirmed the target engagement of the different inhibitors (Appendix A). At these time points, we also observed an increased cleavage of PARP (Appendix A), a substrate of caspase 3 and a hallmark of apoptosis, after treatment with combinations as compared to single agents on MP65 and MM66 cell lines. This increase is higher when VS4718 is combined with LXS196 than with Trametinib. This finding correlates with the higher caspase 3/7 activation levels observed in cells treated with VS4718 and LXS196 when compared to the combination with VS4718 and Trametinib (Figure 2).

### 3.4. In Vivo Evaluation of FAK Inhibitor-Based Combinations in UM PDXs

To evaluate the anti-tumor activity of the two combinations (VS4718 with LXS196 or Trametinib) identified in UM cell lines, we selected three UM PDX models (MM309, MM339, and MM26) obtained from UM metastatic samples displaying GNAQ/11 activating mutations and either BAP1 deficiency (MM309, MM339) or a SF3B1 mutation (MM26). The morphology and molecular characteristics of the selected PDX models are described in Appendix A. The doses, routes of administration, and schedules of drugs (VS4718, LXS196, and Trametinib) are shown in Appendix A. Two different schedules were assessed to evaluate anti-tumor efficacy with an administration four or five days per week every week (continuous schedule) or every two weeks (discontinuous schedule).

At day 18 or 26, as shown in Figure 4, VS4718 alone did not induce a significant decrease in tumor growth in MM309 and MM26, and there was a slight but significant reduction in tumor volume in the model MM339 under the discontinuous and continuous schedules (*p* < 0.05 for both schedules) (Figure 4A,G). At day 18 or 26, treatment with Trametinib alone resulted in a significant decrease in tumor growth in the three models when administered every week (Figure 4D–F). The PKC inhibitor LXS196 significantly reduced tumor growth in the three models and the two schedules, except for the MM309 model under the discontinuous schedule (Figure 4H). As shown in Figure 4, the reduction in tumor volume observed with Trametinib and LXS196 monotherapies was significantly higher with the continuous schedule as compared with the discontinuous one (*p* < 0.04) except for LXS196 in MM339 (Figure 4J).

At day 18 or 26, the combination of Trametinib with VS4718 resulted in a significant reduction of tumor growth in MM339 and MM26 under the continuous schedule, *p* < 0.001 and *p* < 0.03, respectively, and in MM339 under the discontinuous one (*p* = 0.008) when compared to the treatment with the inhibitors alone (Figure 4A–F). We observed toxicity with the combination of these two drugs that forced us to complete the experiments at 18 days for MM339 and MM309 models. The toxicity (loss of weight) happened in mice bearing tumors and was not observed in non-tumor-bearing mice even at doses higher than those used in these experiments (Appendix A).

At day 37 or 26, the combination of LXS196 and VS4718 induced a significant reduction of tumor volume as compared to monotherapies in MM26 and MM339 models under the continuous schedule (*p* = 0.0043) (Figure 4J,L). Importantly, this combination resulted in total tumor regression in 4 out of 6 mice in MM339 (Figure 4J) and 3 out of 5 in MM26 (Figure 4L). This combination was less toxic than that of VS4718 and Trametinib, and the experiments could be pursued for up to 37 days for MM339 and MM309 (Appendix A).

The overall efficacy of the combinations was further analyzed through the probability of progression time taken for tumors to quadruple in size (RTV × 4) and the overall response rate (ORR) considering all the mice included in the experiments (Figure 5). As observed in Figure 5A–D, the combination of VS4718 and LXS196 seems to result in a higher reduction in the probability of tumor progression as compared to the combination of VS4718 and Trametinib, but the differences are not statistically significant (*p* = 0.13).

Under the continuous schedule, the percentage of mice displaying an ORR lower than −0.5 was 20%, 81% and 96% for VS4718, Trametinib or their combination, respectively (Figure 5F). The percentage of mice with an ORR lower than −0.75 was 7%, 24% and 57% when treated with VS4718, Trametinib or their combination (Figure 5F). With this combination, we did not observe any complete regression. Although no significant differences could be observed between the two combinations in ORR lower than −0.5, the impact on ORR values lower than −0.75 of the LXS196 and VS4718 combination were consistently higher as compared to the combination of VS4718 and Trametinib (Figure 5E–H). Importantly, half of the mice included in the experiments displayed a complete regression with VS4718 and LXS196 (7 mice out of 16) (Figure 5H). As expected, the impact of monotherapies or combinations on ORR were significantly lower when drugs were administered under the discontinuous schedule. Finally, an analysis of protein expression of the residual tumors in MM339 PDX treated with inhibitor combinations confirmed target engagement (Appendix A). Overall, our in vivo data confirm that targeting FAK in combination with inhibitors of PKC or MEK represents a valuable therapeutic strategy in UM. 

## 4. Discussion

In recent years, the comprehension of the pathways deregulated in UM has led to the identification of potential therapeutic targets (i.e., PKC, MEK), which have been evaluated both in vitro and in vivo UM models. Unfortunately, single inhibition of these targets has not been translated into a survival benefit for patients in clinical trials. Compensatory mechanisms or cell heterogeneity could result in resistance to the drugs assessed in patients. The toxicity in humans of compounds evaluated as safe in murine models could also be responsible for the absence of efficacy at the permissive doses used in the trials. Combinatorial approaches can be a solution to avoid these issues. It has been demonstrated that the main driver mutations in UM (GNAQ/11 activating mutations) trigger two distinct pathways. Whereas the canonical pathway activates the PLCβ, PKC and MEK/ERK pathway [4], the non-canonical pathway activates YAP in a process controlled by TRIO, RhoA, and FAK [11,15]. Presumably, inhibiting both signaling pathways can result in improved control of malignant cell growth and viability. This has recently been demonstrated targeting FAK and MEK at the same time with selective inhibitors both in vitro and in vivo [17].

In recent years, we have evaluated drug combinations in vitro by using a series of relevant UM cell lines [27]. Given that we did not include FAK inhibitors in these studies, we decided to evaluate combinations of the FAK inhibitor VS4718 with inhibitors targeting other pathways described as deregulated in UM: PKC, MEK, Bcl-2, Bcl-xL, and MDM2. Combinations of VS4718 with the MEK inhibitor Trametinib and the PKC inhibitor LXS196 synergistically reduced the cell viability and induced apoptosis. Importantly, these combinations demonstrated a remarkable in vivo efficacy in UM PDX models. 

When target engagement was assessed in UM cell lines, the inhibitors selectively affected the activation of the expected pathways. While others have shown that PKC inhibition in UM cells affects the phosphorylation of FAK [35], we did not observe any significant reduction of FAK phosphorylation after treatment with LXS196 in any of the assessed UM cell lines. These differences cannot be attributed to the UM cell lines because we also evaluated the same two UM cell lines investigated elsewhere and could not observe a reduction in phospho-FAK levels under LXS196 treatment. 

Ma et al. have reported that the combined inhibition of PKC and MEK but not FAK, and MEK or FAK and PKC synergistically reduces cell viability in UM cells. Whereas this work was done using OMM1.3 and MP41, two BAP1-proficient UM cell lines, the consistent strong synergies we have found by combining a FAK inhibitor with either a MEK inhibitor or a PKC inhibitor were demonstrated by evaluating cell viability and apoptosis induction in 6 different UM cell lines displaying BAP1 deficiency, a mark of poor prognosis in UM. Our data agree with the synergistic effects of the combination of MEK and FAK inhibitors recently reported using a panel of UM cell lines including wildtype and mutant BAP1 cells [17].

Although the efficacy of the two tested combinations was very similar both in vitro and in vivo, the dual inhibition of FAK and PKC has some advantages over the inhibition of FAK and MEK. In vitro, the induction of apoptosis with VS4718 and LXS196 is higher than that obtained with VS4718 and Trametinib. In vivo, the impact on ORR of the LXS196 and VS4718 combination was consistently higher as compared to the combination of VS4718 and Trametinib. Interestingly, half of the mice treated with LXS196 and VS4718 displayed complete tumor regression, which was never observed with the combination of VS4718 and Trametinib. Importantly, a toxicity was observed with VS4718 and Trametinib which prevented follow-up for more than 18 days in two of the models (MM309 and MM339). These data suggest that the inhibition of FAK and PKC may be more suitable for UM patients than the dual inhibition of FAK and MEK. We have observed that in some mice the significant reduction in tumor volume number found with the combination of LXS196 and VS4718 was followed by a re-growth (Figure 4L), suggesting a potential selection of resistant populations. This is a potential limitation for the use of this therapy in humans, which will need to be examined in trials. Future directions of our study will include the comprehension of the mechanisms of resistance displayed by UM cells when treated with PKC and FAK inhibitors.

FAK inhibitors have been assessed in clinical trials alone or in combination with RAF/MEK inhibitors in multiple cancer types, including UM [25,26]. We are not aware of any trial in cancer patients associating a FAK inhibitor with a PKC inhibitor. Our results show that it is worth including this combination in basket clinical trials in UM metastatic patients.

## 5. Conclusions

Synergistic evaluation of the FAK inhibitor with a series of inhibitors targeting UM deregulated pathways in a panel of cell lines showed that the combined inhibition of FAK and MEK or PKC exert remarkable activity in UM patient-derived xenografts. Our study confirms the previously described synergy of the dual inhibition of FAK and MEK, and identifies a novel drug combination (FAK and PKC inhibitors) as a promising strategy for therapeutic intervention in metastatic UM.

## Figures and Tables

**Figure 1 cancers-15-02280-f001:**
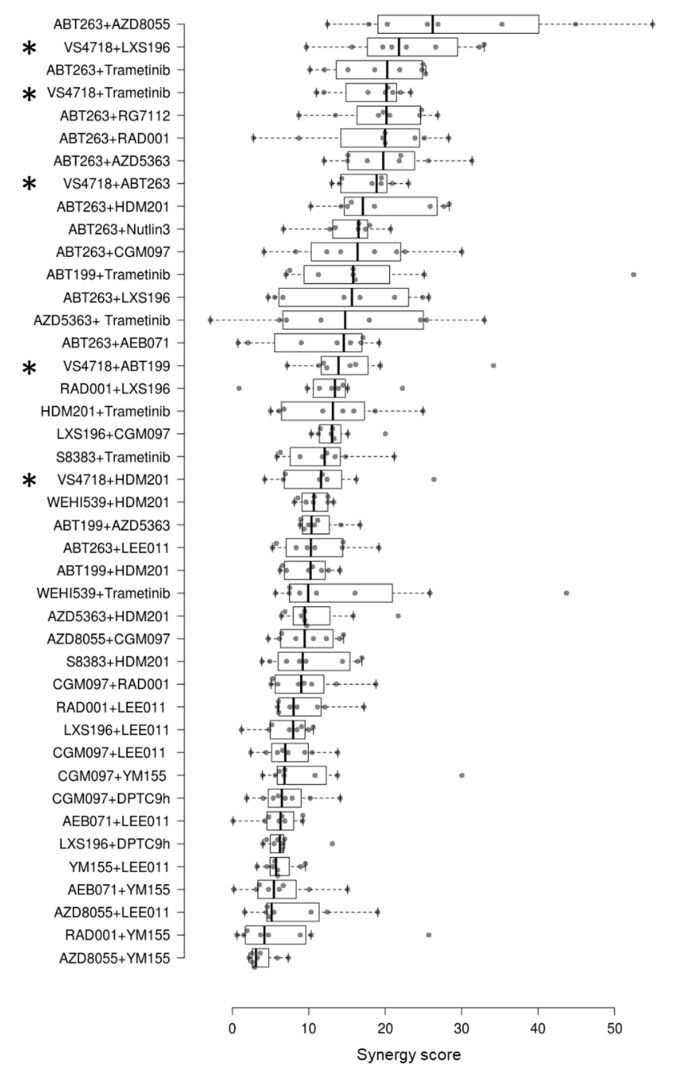
New ranking of drug combinations tested in uveal melanoma cell lines according to synergy scores. The screening of drug combinations was done in eight uveal melanoma cell lines. The FAK inhibitor (VS4718)-based combinations are indicated by asterisks. Each circle represents the synergy score for one cell line. The synergy scores of combinations not including the FAK inhibitor were previously reported in [27].

**Figure 2 cancers-15-02280-f002:**
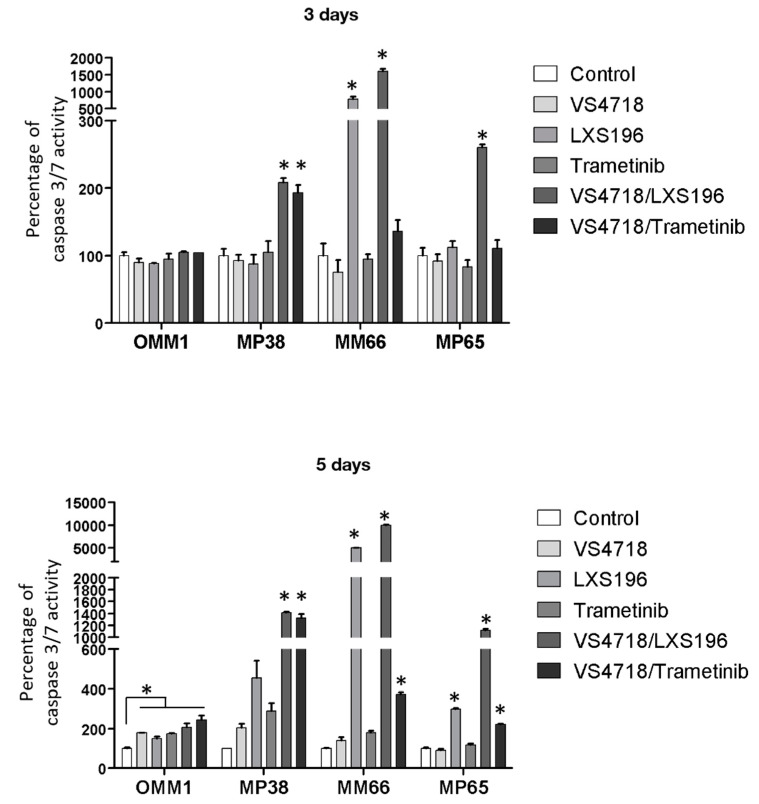
Apoptosis induction in 4 different UM cell lines treated with FAK inhibitor (VS4718) combined with PKC inhibitor (LXS196) or MEK inhibitor (Trametinib). Quantification of apoptosis induction at 3 or 5 days of treatment. A fluorescence-based assay was used to determine the activity of caspase-3 and caspase-7, indicators of apoptosis induction. Data are represented as the mean of triplicates ± SD. Paired *t*-test was used to generate the *p*-values comparing each condition to the control condition; * *p* < 0.05.

**Figure 3 cancers-15-02280-f003:**
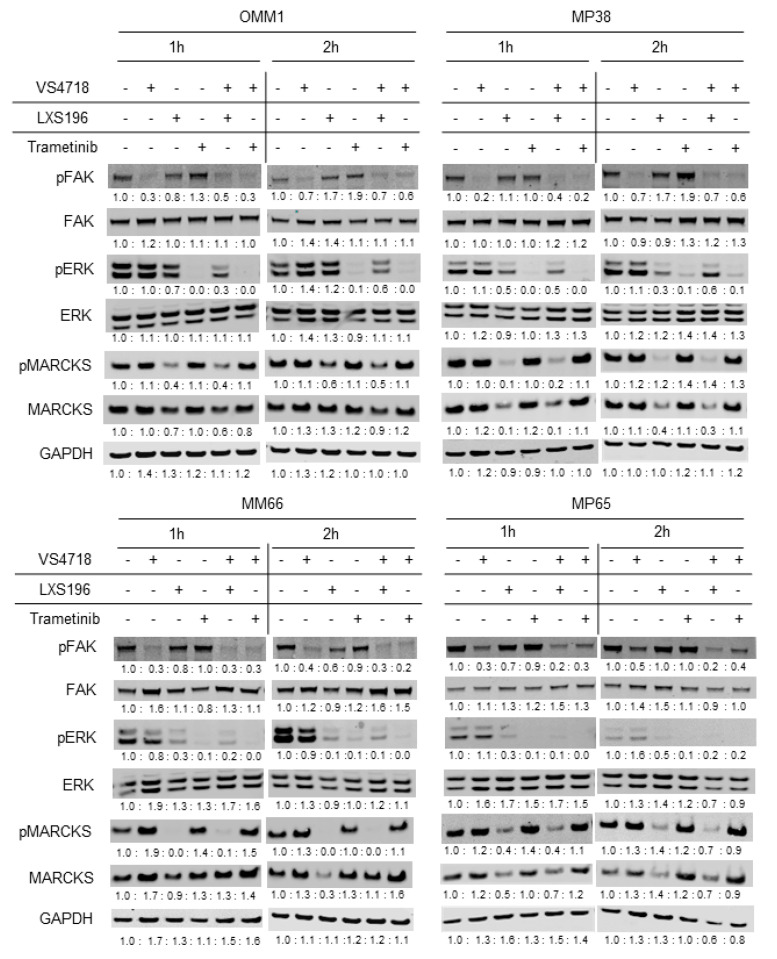
Immunoblot analysis for key signaling pathways in uveal melanoma. Immunoblotting of key signaling pathways in OMM1, MP38, MM66 and MP65 cell lines after 1 and 2 h of treatment with VS4718 (FAK inhibitor) at 1 µM, LXS196 (PKC inhibitor) at 500 nM or Trametinib (MEK inhibitor) at 5 nM alone or in combination. GAPDH is used as a loading control. Intensity readings of each band have been quantified by ImageJ (see Appendix A).

**Figure 4 cancers-15-02280-f004:**
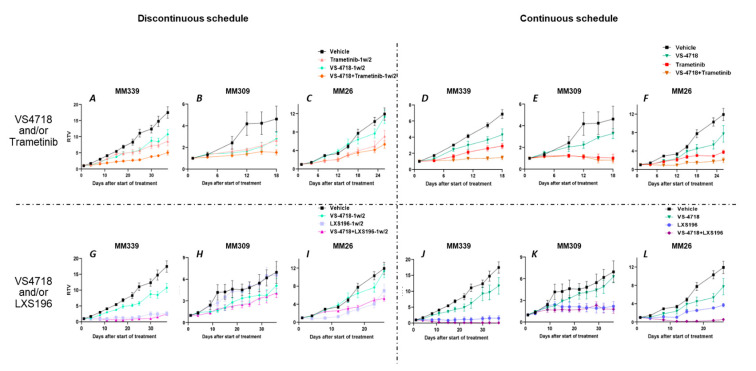
In vivo efficacy assessment: VS4718 combined with Trametinib or LXS196. In vivo efficacy of VS4718 combined with Trametinib (**A**–**F**) or LXS196 (**G**–**L**) in three uveal melanoma patient-derived xenografts according to two schedules of treatment, discontinuous (1/2 w) [**A**–**C** and **G**–**I**] and continuous treatment [**D**–**F** and **J**–**L**]. RTV: relative tumor volume. Mann-Whitney test is applied to evaluate significance.

**Figure 5 cancers-15-02280-f005:**
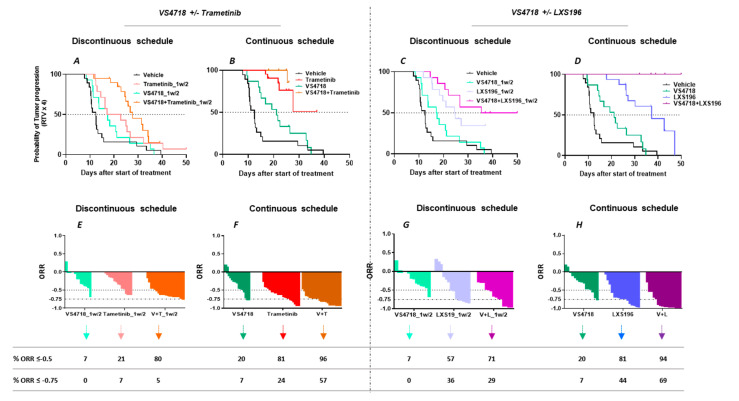
In vivo efficacy assessment: Probability of progression of both monotherapies and combinations (**A**–**D**). The probability of progression after each tested treatment; the methodology is detailed in the Materials & Methods section; the time to reach relative tumor volume (RTV) × 4 for each treated mouse has been calculated. Gehan-Breslow-Wilcoxon test is applied to evaluate significance. ORR of both monotherapies and combinations (**E**–**H**): In vivo efficacy of VS4718 with Trametinib or LXS196 in 3 uveal melanoma patient-derived xenografts. Overall response rate (ORR): the percentage corresponds to the mice with an ORR < −0.5 and ORR < −0.75. Fisher’s test is applied to evaluate significance.

## Data Availability

All data are already shared in the paper.

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
