# Peer review of "FAK Inhibitor-Based Combinations with MEK or PKC Inhibitors Trigger Synergistic Antitumor Effects in Uveal Melanoma"

_cancers, 2023, doi:10.3390/cancers15082280_

Round 1

Reviewer 1 Report

This manuscript describes treatment of uveal melanoma (UM) cell lines and human tumour xenografts with combinations of FAK, MEK and PKC inhibitors, indicating synergistic inhibition of growth for some combinations. Overall, the manuscript is very well written, and figures and tables are well-presented. The study is very preliminary and its limitations should be clearly described. The authors could consider the following comments and suggestions.

1. In Section 3.6, the authors name 3 UM xenografts derived from liver metastases of patients, however there are no references to the origin or derivation of these models, their source or consent/ethics approval for their establishment. Could these details please be added.

2. On the first line of page 4, the authors mention an experimental protocol for the in vivo studies in ‘supplementary methods’, which do not appear to have been uploaded. Without these details, it has not been possible to review this part of the manuscript, in particular the discontinuous/continuous dosing of mice.

3. Also on page 4, lines 156-157, were tumour volumes measured manually or using imaging? This should be stated.

4. There are insufficient details of the drugs and drug combinations tested in relation to results presented in Figure 1. In light of results presented in this figure, why did the authors focus on the FAKi combinations when other combinations appear to be more active? Details of Methods, Results and results interpretation in relation to this figure should be added to the manuscript.

5. What were the drug-induced toxicities observed in these mice (page 8, final 2 paragraphs)? Were the toxicities only observed in tumour-bearing mice or were they solely drug-related (i.e. observed in non-tumour bearing mice)? Would such drug toxicities prevent treatment of human patients? Were experiments repeated with lower drug doses - did the drug combinations have anti-tumour activity at non-toxic doses. Further details should be provided.

6. With regards to the tumour xenograft model, what happened when drug treatments were stopped? Did tumours regrow? The authors mention “total tumour regression” in a number of mice. On what basis did they arrive at this conclusion? How long after cessation of drug treatments were mice kept to confirm that the tumours had completely regressed? Was this supported by pathological confirmation (necropsy)?

7. What information can the authors provide regarding the remaining xenografts following drug treatments? At the end of experiments, were the remaining tumours necrotic or apoptotic? Was cell proliferation still present (e.g. mitotic figures)? Although providing interesting preliminary observations, is there any indication of the future success of these types of drug combinations or do the authors feel that the short term growth inhibition in the in vivo experiments is rapidly leading to selection of resistant populations (the study largely simply shows tumour (xenograft) growth inhibition, not tumour regression).

8. Because this study is quite preliminary and observational, discussion of the limitations and future directions of the work would be helpful.

Reviewer 2 Report

This review proposes that FAK inhibitor-based combinations with MEK or PKC inhibitors trigger synergistic antitumor effects in uveal melanoma. The current manuscript has some study that need to be addressed. The authors also need to show this is due to on-target activity of the combination.

There are some of the studies that can be considered.

1) The authors could also include immunoblots of cleaved PARP and cleaved caspase 3 to show that combinations play a role in apoptosis in-vitro.

2) Currently, in vivo pharmacodynamics data is lacking. Does FAK inhibitor-based combinations with MEK or PKC inhibitors suppress FAK and MEK or PKC in vivo, like it does in vitro?

3) Mice body weight could also be included to show that (FAK and PKC) inhibitors did not show much toxicity

The findings of the studies documented in this manuscript are interesting and potentially relevant to

disease treatment

Round 2

Reviewer 1 Report

I feel that the investigators have satisfactorily answered reviewers’ comments and that the manuscript is suitable for publication. There are minor grammatical errors in the text that require correction and there seems to be a formatting error in Figure 3 and the Figure 3 legend.